# Antibacterial Effectiveness of Fecal Water and In Vitro Activity of a Multi-Strain Probiotic Formulation against Multi-Drug Resistant Microorganisms

**DOI:** 10.3390/microorganisms8030332

**Published:** 2020-02-27

**Authors:** Alessandra Oliva, Maria Claudia Miele, Massimiliano De Angelis, Silvia Costantini, Maria Teresa Mascellino, Claudio Maria Mastroianni, Vincenzo Vullo, Gabriella d’Ettorre

**Affiliations:** Department of Public Health and Infectious Diseases, Sapienza University of Rome, 500185 Piazzale Aldo Moro, Italy; mariaclaudia.miele@uniroma1.it (M.C.M.); massimiliano.deangelis@uniroma1.it (M.D.A.); costantini.1405865@studenti.uniroma1.it (S.C.); mariateresa.mascellino@uniroma1.it (M.T.M.); claudio.mastroianni@uniroma1.it (C.M.M.); vincenzo.vullo@uniroma1.it (V.V.); gabriella.dettorre@uniroma1.it (G.d.)

**Keywords:** probiotics, multi-drug resistant microorganisms, antibacterial activity, gut colonization, decolonization

## Abstract

**Introduction:** Intestinal colonization with multi-drug resistant (MDR) microorganisms is a consequence of antimicrobial-induced gut dysbiosis. Given the effect of probiotics in modulating gut microbiota, the aim of the study was to investigate whether the ingestion of high concentration multi-strain probiotic formulation would change the antibacterial activity of the feces against clinical strains of MDR microorganisms. The corresponding in vitro antibacterial activity was also investigated. **Materials/Methods:** The feces of healthy donors (*n* = 6) were analyzed before and after a 7-day dietary supplementation with a multi-strain probiotic formulation and tested against MDR microorganisms of clinical concern (carbapenem-resistant (CR), *Klebsiella pneumoniae* (CR-Kp), CR-*Acinetobacter baumannii* (CR-Ab), CR-*Pseudomonas aeruginosa* (CR-Pa), and methicillin-resistant *Staphylococcus aureus* (MRSA)). The tested MDR pathogens were cultured with decreasing concentrations of fecal water obtained before and after the treatment period. Furthermore, to corroborate the results obtained from the feces of healthy donors, the in vitro antibacterial activity of probiotic formulation (both whole probiotic (WP) and probiotic surnatant (PS)) against the same collection of MDR microorganisms was evaluated at different incubation times throughout the minimum bactericidal dilution and the corresponding serial silution number. **Results:** While before probiotic administration, the fecal water samples did not inhibit MDR microorganism growth, after supplementation, a reduced bacterial growth was shown. Accordingly, a noticeable in vitro activity of WP and PS was observed. **Conclusions:** Although preliminary, these experiments demonstrated that a specific multi-strain probiotic formulation exhibits in vitro antibacterial activity against MDR microorganisms of clinical concern. If confirmed, these results may justify the administration of probiotics as a decolonization strategy against MDR microorganisms.

## 1. Introduction

The intestinal microbiota of healthy humans is primarily comprised of a variety of aerobic-tolerant and anaerobic genera, including *Lactobacillus* spp, *Faecalibacterium* spp, *Bifidobacterium* spp, and *Bacteroides* spp, in a mutualistic and homeostatic relationship with the host [1]. When, for a number of reasons—e.g., drug treatments, irradiation, alcohol consumption, early-life antibiotic exposure, prolonged antimicrobial treatment, and hospitalization—such equilibrium is affected, *Escherichia coli*, *Klebsiella pneumoniae,* or *Proteus* spp, and other potentially pathogenic genera may increase their presence in the gut [2] and even become multi-drug resistant (MDR) [3]. MDR microorganisms are difficult to eradicate and are responsible for treatment failures, prolonged hospitalizations, and high mortality rates [4].

Being that the gut is considered the main reservoir of MDR microorganisms, gut decolonization has been proposed and antibiotics are the drugs of choice for this purpose. However, the trials mainly conducted with gentamicin or colistin have generated conflicting results and, therefore, doubts about the real value of this approach, especially in the long-term [5,6].

According to the Aristotelian principle that “there are no vacuums in nature,” which could explain why bactericidal antibiotics are not as effective as expected, the administration of bacteria with beneficial properties (probiotics) to “occupy and/or compete for the space” with MDR microorganisms may be a safe alternative to antibiotics or a complement to the antibacterial treatment [7,8]. The studies performed so far with probiotics have produced confusing results [9,10], but this is not surprising, since most included the administration of a single (i.e., *Lactobacillus ramnhosus*) or few bacterial species, and the selection of the probiotic products was routinely based on the in vitro results. In fact, the common in vitro testing systems (agar disk diffusion, agar dilution, and broth dilution) do not include all the biological variables found in the human body that can affect the probiotic capability to survive, proliferate, and release molecules with antimicrobial activity. The in vitro evaluation of the minimum inhibitory concentration (MIC) of probiotics does not take into account the variability of the probiotic distribution on the gut mucosal surfaces, of its critical mass in the fecal material, or of the induced changes in pH at different anatomic sites. Last but not least, the in vitro testing systems do not consider that the growth of the pathogen may be suppressed without the release of substances endowed with anti-microbial effects by the probiotic bacteria because they interfere with the nutrients or modulate the host’s immune system and inflammatory pathways.

So as not to rely only on the in vitro antibacterial tests, we selected a probiotic formulation that could be a candidate for preventing MDR infection in the patients at risk by evaluating its antimicrobial activity in the feces of healthy donors (HD, *n* = 6) before and after one week of supplementation with the product. We also compared the ex vivo results with the in vitro antimicrobial activity when the same probiotic was tested against clinical strains of MDR Gram-negative bacilli and methicillin-resistant *Staphylococcus aureus* (MRSA).

## 2. Materials and Methods

### 2.1. Demographic Characteristics of the Participants

The present study is a pilot study that includes healthy individuals (*n* = 6). All subjects were enrolled at the Clinic of the Department of Public Health and Infectious Diseases of the University of Rome “Sapienza.” The study protocol was approved by the internal committee of the Department of Public Health and Infectious Diseases of “Sapienza,” University of Rome, and by the Ethics Committee of Policlinico Umberto I Hospital, Rome (2970). All participants agreed by signing a written informed consent form to be treated for a week with a probiotic. For each subject, two samples of stools were collected: one before (day 0, D0) and one after the supplementation (day 7, D7). All study participants were healthy Caucasian men (health donors—HDs) with a median age of 35 years (interquartile range (IQR) 22–53 years). The exclusion criteria were: i) a known allergy or intolerance to the product; ii) diarrhea; iii) a history of or current inflammatory diseases of the small or large intestine; iv) a previous or current drug addiction; v) past gut infections. All subjects enrolled in the study showed a negative microscopic examination of the stool, a negative bacterial stool culture, and a negative result for the research of pathogenic viruses in the stool. For all donors, the supplementation dose was three sachets per day for a total of 1350 billion bacteria per day for 7 days. This dosage is in the range of the daily amount recommended by the marketers and previously reported in the literature [9].

### 2.2. Probiotic Formulation

For the ex vivo and in vitro experiments, a multi-strain probiotic formulation (trade names: Vivomixx® in EU, Visbiome® in USA) containing 450 billion bacteria of *Lactobacillus paracasei* DSM 24,734, *Lactobacillus plantarum* DSM 24,730, *Lactobacillus acidophilus* DSM 24,735, *Lactobacillus delbruckei subspecies bulgaricus* DSM 24,734, *Bifidobacterium longum* DSM 24,736, *Bifidibacterium infantis* DSM 24,737, *Bifidobacterium breve* DSM 24,732, and *Streptococcus thermophilus* DSM 24,731 was used. To permit the optimal growth of *Lactobacillus* spp, *S. thermophilus*, and *Bifidobacteria*, the M.R.S. (De Man, Rogosa e Share) with the addition of 0.5 gr/L of cysteine broth was used for all in vitro experiments [10].

### 2.3. Tested Strains

The in vitro activity of the probiotic formulation was investigated against clinical strains of carbapenem-resistant (CR) *K. pneumoniae* (CR-Kp), CR-*Acinetobacter baumannii* (CR-Ab), CR-*Pseudomonas aeruginosa* (CR-Pa), and MRSA. After bacterial storage on a cryovial bead preservation system (Microbank; Pro-Lab Diagnostics, Richmond Hill, ON, Canada) at –80 °C, the inoculum was prepared by spreading one cryovial bead on blood agar plate and incubating overnight at 37 °C. One colony was resuspended in 3 mL of NaCl and then was adjusted to a turbidity of 0.5 McFarland, corresponding to ~1.5 × 10^8^ CFU/mL.

### 2.4. Anti-MDR Microorganism Activity of Fecal Water

The stools (200 mg) collected at day 0 and day 7 (before and at the end of the 7-day probiotic supplementation, respectively) were dissolved in 2 mL of phosphate buffered saline (PBS), with a final concentration of 100 mg/mL, vortexed, and then assayed for anti-MDR activity (fecal water). A final MDR inoculum of 5 × 10^5^ CFU/mL was added to tubes containing 2-fold serial dilutions of the Mueller Hinton broth (MHB) plus fecal water at the final concentration of 50 mg/mL and incubated at 37 °C for 24 h. A total of 50 μL was then plated on the Mueller Hinton Agar (MHA) and the colonies were counted after an additional 24 h of incubation. Minimal bactericidal dilution (MBD) was defined as the fecal water dilution obtaining ≥99.9% (i.e., ≥3log10 CFU/mL) reduction of the initial MDR count after 24 h of incubation, whereas the serial dilution number (SDN) corresponded to the number of the serial 2-fold probiotic dilutions needed to obtain the corresponding MBD [11]. The limit of detection was 20 CFU/mL. Each experiment was run in duplicate. The data were expressed as mean ± standard deviation (SD).

### 2.5. Antibacterial Activity of Probiotic Formulations

For each in vitro experiment, the aforementioned probiotic (P) formulation was used as whole (WP) and cell-free surnatant (PS) formulations, respectively. Different probiotic incubation times (4, 24, 48, and 72 h) were considered.

#### 2.5.1. Anti-Bacterial Activity of WP and PS 

For the analysis of the WP, the dried powder of the probiotic formulation was dissolved in 20 mL of MRS (Man, Rogosa and Sharpe) broth. Afterwards, 450 µL of the obtained solution were added to an equal volume of MRS broth with a final volume of 900 µL (1:2 dilution), and serial 2-fold dilutions (up to 1:256) of probiotic formulation were then performed. A final inoculum of ~5 × 10^5^ CFU/mL of CR-Kp, CR-Ab, CR-Pa, and MRSA was then added to the tubes containing 2-fold serial dilutions of the probiotic solution and incubated at 37 °C in aerobic conditions for 24 h.

To obtain the PS, the resuspended probiotic (obtained as previously described) was centrifugated at 3000 rpm for 15 min, and the cell-free surnatant was collected [12] and further incubated at 37 °C in aerobic conditions for 24 h.

After the incubation of WP and PS, aliquots of 50 µL were plated on blood agar plates and the numbers of bacteria were determined. The MBD (minimum bactericidal dilution) was defined as the lowest dilution that induced ≥99.9% (≥3-log10 CFU/mL) reduction of the initial bacterial count after 24 h of incubation, and the serial dilution number (SDN) corresponded to the number of the serial 2-fold WP and PS dilutions needed to obtain the corresponding MBD. The limit of detection was 20 CFU/mL.

Following the observed high activity of the original probiotic formulation, we conducted further experiments to assess whether WP and PS maintained the antibacterial activity after incubation at different times (4, 24, 48, and 72 h) [13]. The probiotic formulations and the serial 2-fold dilutions were prepared as previously described.

To lower the possible influence of the acid compounds produced by the fermenting activity of probiotic bacteria on the WP and PS activity, the pH solution was checked at each time point and adjusted with NaOH in order to obtain 6.5–7.5 values.

#### 2.5.2. Anti-Bacterial Activity of WP and PS Following 4 h and 24 h of Co-Incubation with the Pathogen

In addition to the probiotic formulation alone, we wondered whether the co-incubation with the same pathogen (i.e., co-incubation with CR-Kp to test the antibacterial activity against CR-Kp) could have an influence on the antibacterial activity of the cocktails of probiotics.

Thus, a total of four dried probiotic formulations were used, dissolved in MRS broth as previously described, and further co-incubated at 37 °C for 4 h and 24 h in aerobic conditions with a final inoculum of 1 × 10^5^ CFU/mL of CR-Pa, CR-Ab, CR-Kp, and MRSA.

Following the 4 h and 24 h co-incubation, the activity of WP and PS (obtained as above) against the same co-incubated pathogen was evaluated. The Ph of the solution was checked and, if needed, adjusted with NaOH to reach 6.5–7.5 values.

The antibacterial activity was expressed as a MBD (obtained as above) and as a serial dilution number (SDN), which corresponded to the number of the serial 2-fold WP and PS dilutions needed to obtain the corresponding MBD.

All in vitro experiments were performed in duplicate. The results were expressed as mean (± standard deviation (SD)).

## 3. Results

### 3.1. Anti-MDR Microorganism Activity of Fecal Water

Before probiotic administration, the fecal water samples of all volunteers (*n* = 6) did not show antibacterial activity, with the presence of MDR microorganism growth even at the lowest 2-fold serial dilution (1:2, SDN 1). Conversely, after 7 days of probiotic supplementation, the fecal water samples showed bactericidal activity at higher dilutions, especially against CR-Pa and CR-Ab (Figure 1).

### 3.2. Anti-Bacterial Activity of WP and PS 

Overall, WP and PS showed a high antibacterial activity against all the tested strains, especially against the non-fermenting Gram-negative bacteria CR-Ab and CR-Pa. In particular, for WP, the MBDs were 1:256 (SDN 8), 1:128 (SDN 7), 1:32 (SDN 5), 1:16 (SDN 4) for CR-Pa, CR-Ab, MRSA, and CR-Kp, respectively. This was slightly higher than PS, which showed MBDs of 1:32 (SDN 5), 1:16 (SDN 4), 1:8 (SDN 3), and 1:8 (SDN 3) for CR-Pa, CR-Ab, MRSA, and CR-Kp, respectively (Figure 2).

In line with the results obtained at T0, the antibacterial activity of WP was higher than PS and especially noticeable against CR-Pa and CR-Ab. Following an incubation of 4 h, the MBDs of WP were 1:64 (SDN 6), 1:64 (SDN 6), 1:32 (SDN 5), 1:8 (SDN 3) vs. 1:16 (SDN 4), 1:16 (SDN 4), 1:8 (SDN 3), 1:4 (SDN 2) of PS for CR-Pa, CR-Ab, CR-Kp, and MRSA, respectively (Figure 3A). When incubating the WP and PS for longer, a lower antibacterial activity was overall observed, with a plateau obtained at 48 h and maintained up to 72 h (Figure 3A,B).

### 3.3. Anti-Bacterial Activity of WP and PS Following 4 and 24 h of Co-Incubation with the Pathogen

Following a 4 h co-incubation between the MDR microorganism pathogens and the probiotic formulation, the MBDs of WP and PS were 1:128 (SDN 7), 1:128 (SDN 7), 1:64 (SDN 6), and 1:16 (SDN 4) for WP, and 1:8 (SDN 3), 1:4 (SDN 2), 1:4 (SDN 2), 1:8 (SDN 3) for PS against CR-Pa, CR-Ab, CR-Kp, and MRSA, respectively (Figure 3A–D). Following the 24 h of co-incubation, the antibacterial activity was consistently lower for CR-Pa and MRSA and absent for CR-Ab and CR-Kp (Figure 4A–D).

The comparison between the antibacterial activity at 4 h and 24 h under different experimental conditions is shown in Table 1.

## 4. Discussion

On average, a person eliminates approximately 200 gr of fresh feces consisting of 75% water and 25% solid matter. About 30% (15 g) of the solid matter consists of live and dead bacteria and the rest is indigestible food matter such as cellulose (30%), cholesterol and other fats (10–20%), inorganic substances such as calcium phosphate and iron phosphate (10–20%), and protein (2–3%). The extracted liquid portion of whole feces is termed fecal water and has been used in our study as an indicator for the presence of microbial substances that are antagonistic to the growth of other microbes in the gut. While a number of physiological factors reduce the availability of antibiotics and what is measured in the feces is a fraction of the unchanged active principle of the drug that is excreted, for probiotics, the scenario is different. If the probiotic bacteria are able to compete with the intestinal flora and colonize the gut, what will be measured in the feces is an amplification of their properties, including the possible production of active substances following their replication, and the interactions with the flora and the immune system of the host. In other words, while for antibiotics, the fecal antimicrobial effect is the result of a series of physiological factors influencing the concentration of the drug (absorption, enzymatic inactivation, protein binding, liver metabolism, etc.), for the probiotic products, the fecal antimicrobial effect is the summation of various local gut microbial factors (expansion or reduction of selected species, production of metabolites, etc.). Starting from the above assumption, it is explained why the in vitro tests for antibiotics are always more “brilliant” than the results obtained in vivo. On the contrary, for probiotics, considering the interaction of these live organisms in the gut, it may be possible that in vivo, they are far more effective than in vitro, where only their capability to produce bacteriocins is taken into consideration [14].

In a previous study, we have shown that the feces from healthy subjects do not affect the growth of *Candida albicans*, but depending on the probiotic formulation administered, a 7-day treatment increased the fecal alfa interferon and the fecal anti-*Candida* activity [11].

In the present study, the same probiotic formulation that was found to be effective for increasing the fecal anti-*Candida* activity was tested against clinical strains of MDR Gram-negative bacilli and MRSA, which are well-known colonizers of the gut of hospitalized and pluri-treated subjects, and whose presence in the intestine is a risk factor for subsequent infections.

Similar to the anti-fungal activity, we have observed that before probiotic administration, the feces from healthy subjects did not influence bacterial growth, whereas following a 7-day course of probiotic supplementation, a noticeable anti-bacterial activity of the fecal water samples was indeed observed.

The results of this study may have important clinical consequences in terms of selecting “the right probiotic formulation” and, eventually, defining a probiotic-based decolonization strategy in the presence of intestinal MDR microorganism colonization in subjects at high risk of developing MDR microorganism systemic infections, with or without antibiotics. According to a recent article, when the subjects were given antibiotics and treated with a probiotic preparation, the probiotic colonization delayed the recovery of the host’s microbiota for six months [15]. If we consider that MDR microorganisms are usual “guests” in the hospitals and patients’ gut flora, a course of probiotics may protect the hospitalized patients for several months from acquiring MDR germs.

However, we should take into consideration that a recent randomized control trial study did not find a successful long term eradication of extended-spectrum beta lactamases (ESBL), producing microorganisms in the gut of subjects receiving probiotics in comparison with a placebo [9]. Although the conclusion of the study was not to recommend probiotics as an ESBL decolonization strategy, the authors acknowledged that the study was underpowered and the results possibly influenced by the small sample size, rendering the conduction of further investigations necessary [9,16,17].

Since one of the major limitations of the present study resides on the lack of dynamic chemical analyses of the substances present in the feces and induced by the probiotic formulation, we also performed a number of in vitro experiments.

From our combined ex vivo and in vitro results, we found a noticeable activity of the probiotic formulation against a collection of clinical strains of MDR pathogens usually representing the “normal” flora of hospitalized patients.

It is worth noting that the antibacterial activity was dependent on the different statuses of the probiotic formulation, meaning that the whole compound exhibited a greater bactericidal activity compared to what was observed with the cell-free supernatant only.

To the best of our knowledge, no studies comparing the antibacterial activity of the cell-containing and cell-free probiotic formulations against MDR microorganisms have been performed so far, thus highlighting the strength of the present investigation.

The increased action of the whole compound against pathogens may suggest that it is the interaction between live microorganisms rather than only the production of antibacterial substances mainly present in the surnatant that is responsible for the observed antibacterial properties. Furthermore, it was shown that the interaction between the pathogens and the probiotic compared with the sole probiotic formulation was able to stimulate the latter to produce specific substances that increase the killing capacity (i.e., bacteriocins) [18]. This is, in fact, confirmed by the co-incubation experiments, which showed a more potent antibacterial activity of the probiotic formulation after a short incubation with the same pathogen than of that observed without the co-incubation. These findings highlight that the simultaneous presence of probiotics and pathogens may be required in order to augment the antibacterial activity of the compound and stimulate the probiotic mix to produce specific molecules in response to the single pathogen (i.e., antimicrobial peptides and AMPs, such as bacteriocins).

However, a more prolonged co-incubation did not exhibit the same potent antibacterial activity, as shown in the 24-h co-incubation experiments. This is possibly due to either the production of toxic substances during the longer co-incubation or to the development of a kind of antagonism between live microorganisms (probiotics and pathogens) growing in the same medium for 24 h.

The same result was obtained when the whole compound and the cell-free surnatant were incubated for up to 72 h, with the highest antibacterial effect observed without incubation, or after 4 h of incubation and with a lower, more stable effect after longer incubation times. This is an interesting result, since the original formulation contains live bacteria and, when reconstituted for the in vitro experiments or following a short incubation (4 h), it induces an interaction among live microorganisms, the produced antibacterial substances, and the tested pathogens, which is possibly responsible for the observed high antibacterial properties.

Again, the reduction of the antibacterial activity with prolonged incubation may be explained by the in vitro experimental conditions (pathogens and probiotic bacteria growing in the same medium for up to 72 h), which may have induced the production of toxic substances derived from the metabolism of microorganisms.

This factor should be taken into consideration when interpreting the results of in vitro experiments, which may be influenced by several experimental conditions and, therefore, may not truly reflect the in vivo effect of the tested substances, suggesting that in vivo, the effect of the probiotic formulation in reducing bacterial growth may be maintained over time, as described in the literature [15].

An interesting finding of the present study is that non-fermenting Gram-negative pathogens (i.e., CR-Pa, CR-Ab) were found to be more sensitive to the action of the probiotic mix, while the bactericidal capacity of the probiotic formulation against the fermenting Gram-negative strains (i.e., CR-Kp) was found to be slightly lower. No data possibly accounting for this effect are present in the literature; however, it may be hypothesized that the presence of a thick layer of glycocalyx that surrounds CR-Kp and not CR-Pa or CR-Ab makes it more resistant to the action of the various components that are secreted by the probiotic mix.

To corroborate the findings of the present investigation, during incubation, the pH of the probiotic formulation was continuously corrected to values close to 6.5–7.5, so that the possible antibacterial action of lactic acid, which is the final product of the bacterial fermentation process, has been excluded [19]. In fact, other substances produced in the metabolic process of probiotics (i.e., short chain fatty acids (SCFAs), such as propionic, acetic, butyric, and isobutyric acids) may exhibit toxic effects on bacteria at high concentrations. Several in vitro studies have demonstrated that the bacterial toxicity was attributable to the non-ionized forms of these acids into the bacterial cytoplasm [20,21]. These non-ionized forms are small and uncharged and, therefore, are thought to freely diffuse across the bacterial membrane. Once inside the bacterial cytoplasm, which generally has a circumneutral pH, these non-ionized acids dissociate, leading to an accumulation of protons and SCFA anions [22]. On one hand, the influx of protons acidifies the intracellular compartment, ultimately compromising bacterial metabolic reactions and energy conservation; on the other hand, the accumulation of SCFA anions in the cytoplasm may significantly impact cellular physiology by altering the osmotic balance [23].

Therefore, it may be hypothesized that the observed high antibacterial activity of the probiotic formulation resides on the production of SCFAs obtained as a consequence of probiotic metabolism, which is augmented by a short co-incubation with the pathogen. Recently, SCFAs have gained interest as post-biotic compounds, which refers to the soluble factors secreted by live bacteria or released after bacterial lysis, possibly offering physiological benefits to the host [24,25]. 

From a clinical standpoint, our findings strengthen the difference between the use of a simple decolonization strategy based on the administration of oral antibiotics and a complex strategy based on probiotic supplementation, which takes into account the interaction between the probiotic itself, the gut resident flora, and the host’s immune and intestinal cell. Therefore, the results of the present investigation may represent the starting point for future studies investigating the role of the probiotic for gut MDR microorganism decolonization [26,27].

This pilot study has several limitations, such as: i) the low number of subjects, which may impede to derive definitive conclusions; ii) the short experimental period of probiotic supplementation (7 days); and iii) the absence of the analysis of fecal water activity against MDR microorganisms for longer after probiotic supplementation.

## 5. Conclusions

In conclusion, it was herein demonstrated that: i) a 7-day probiotic supplementation increases the fecal activity against MDR microorganisms; ii) a multi-strain probiotic formulation exhibits high in vitro antibacterial activity against MDR microorganisms of clinical concern; iii) the higher activity of the whole formulation than the cell-free surnatant might be due to the interaction between live microorganisms rather than only to the production of antibacterial substances exclusively present in the cell-free surnatant.

If confirmed in the future, the results of the present investigation support the potential use of this specific multi-strain probiotic formulation as an antimicrobial strategy for the treatment of intestinal colonization caused by MDR microorganisms.

## Figures and Tables

**Figure 1 microorganisms-08-00332-f001:**
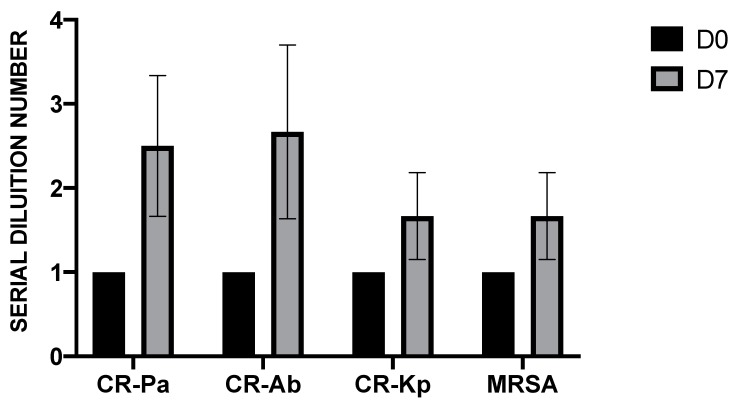
Antibacterial activity of fecal water samples collected from six healthy donors before (day 0, D0) and after (day 7, D7) 7 days of probiotic supplementation against clinical strains of carbapenem-resistant (CR) *P. aeruginosa* (Pa), CR *A. baumannii* (Ab), CR *K. pneumoniae* (Kp), and methicillin-resistant *S. aureus* (MRSA). The serial dilution number (y axis) corresponds to the number of the serial 2-fold fecal water dilutions needed to obtain the corresponding minimal bactericidal dilution (MBD). Values are represented as mean ± standard deviation (SD).

**Figure 2 microorganisms-08-00332-f002:**
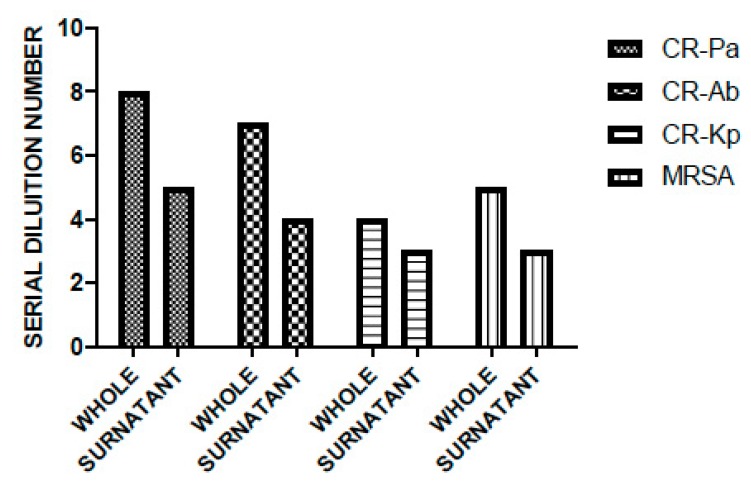
Antibacterial activity of probiotic whole compound (WP) and probiotic cell-free surnatant (PS) against clinical strains of carbapenem-resistant (CR) *P. aeruginosa* (Pa), CR *A. baumannii* (Ab), CR *K. pneumoniae* (Kp), and methicillin-resistant *S. aureus* (MRSA). The serial dilution number (y axis) corresponds to the number of the serial WP and PS dilutions needed to obtain the corresponding minimal bactericidal dilution (MBD). Results were expressed as mean. Standard deviation (SD) is not graphically represented, since the values of the experiments (performed in duplicate) were the same.

**Figure 3 microorganisms-08-00332-f003:**
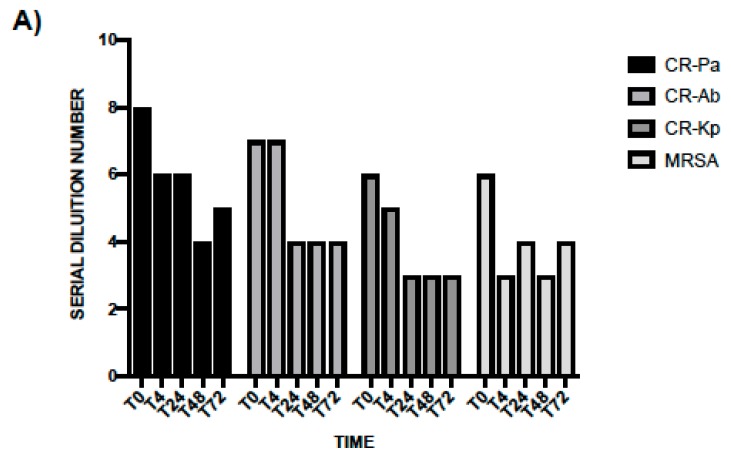
Antibacterial activity of probiotic whole compound (WP, **A**) and probiotic cell-free surnatant (PS, **B**) against clinical strains of carbapenem-resistant (CR) *P. aeruginosa* (Pa), CR *A. baumannii* (Ab), CR *K. pneumoniae* (Kp), and methicillin-resistant *S. aureus* (MRSA) at different incubation times. The serial dilution number (y axis) corresponds to the number of the serial WP and PS dilutions needed to obtain the corresponding minimal bactericidal dilution (MBD). Results were expressed as mean. Standard deviation (SD) is not graphically represented since the values of the experiments (performed in duplicate) were the same.

**Figure 4 microorganisms-08-00332-f004:**
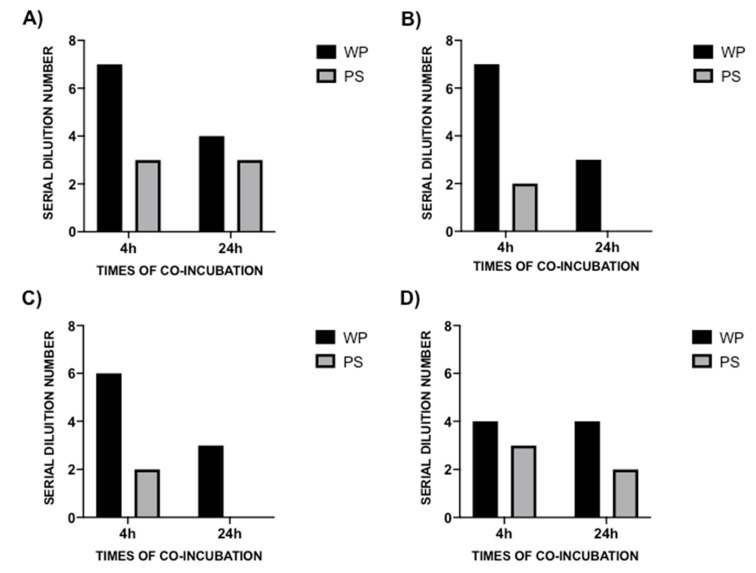
Antibacterial activity of probiotic whole compound (WP) and probiotic cell-free surnatant (PS) against clinical strains of carbapenem-resistant (CR) *P. aeruginosa* (Pa, **A**), CR *A. baumannii* (Ab, **B**), CR *K. pneumoniae* (Kp, **C**), and methicillin-resistant *S. aureus* (MRSA, **D**) at 4 h (black blocks) and 24 h (gray blocks) co-incubation times. The serial dilution number (y axis) corresponds to the number of the serial WP and PS dilutions needed to obtain the corresponding minimal bactericidal dilution (MBD). Results were expressed as mean. Standard deviation (SD) is not graphically represented since the values of the experiments (performed in duplicate) were the same.

**Table 1 microorganisms-08-00332-t001:** Antibacterial activity of probiotic whole compound (WP) and probiotic cell-free surnatant (PS), alone and after co-incubation with the pathogen at 4 h (T4) and 24 h (T24) against clinical strains of carbapenem-resistant (CR) *P. aeruginosa* (Pa), CR *A. baumannii* (Ab), CR *K. pneumoniae* (Kp), and methicillin-resistant *S. aureus* (MRSA) with and without co-incubation with the pathogen. MBD = minimal bactericidal dilution. SDN = Serial Dilution Number (number of the serial 2-fold WP and PS dilutions needed to obtain the corresponding MBD). WP-C = WP after co-incubation; PS-C = PS after co-incubation; NE = no effect. Results were expressed as mean. Standard deviation (SD) is not represented since the values of the experiments (performed in duplicate) were the same.

*Pathogens*	MBD (SDN) (T4)				MBD (SDN) (T24)			
	WP	PS	WP-C	PS-C	WP	PS	WP-C	PS-C
CR-Pa	1:64 (6)	1:16 (4)	1:128 (7)	1:8 (3)	1:64 (6)	1:16 (4)	1:16 (4)	1:4 (2)
CR-Ab	1:128 (7)	1:16 (4)	1:128 (7)	1:4 (2)	1:16 (4)	1:8 (3)	1:8 (3)	NE
CR-Kp	1:32 (5)	1:8 (3)	1:64 (6)	1:4 (2)	1:8 (3)	1:8 (3)	1:8 (3)	NE
MRSA	1:8 (3)	1:4 (2)	1:16 (4)	1:8 (3)	1:16 (4)	1:4 (2)	1:32 (5)	1:4 (2)

## Data Availability

The data used to support the findings of this study are available from the corresponding author upon request.

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
