# Peer review of "Antibacterial Effectiveness of Fecal Water and In Vitro Activity of a Multi-Strain Probiotic Formulation against Multi-Drug Resistant Microorganisms"

_microorganisms, 2020, doi:10.3390/microorganisms8030332_

Round 1

Reviewer 1 Report

The repeat statements on the antibacterial activity should be improved. The definition of SI4 and LI24...is not necessary in the text. The person number in fecal test should be defined. The original faeces showed a 2 to the 1 order activity, is it true it is without mean value? It is suggested that the antibacterial activity being expressed in mean values which might not be an integer. It seems that the longer incubation to 24h is not good for expression of antibacterial activity, this means increased cells biomass  is bad for activity. Any comment? The same case that the original cell  formulation is the best in activity, the incubation decreased its activity, why? The conclusion 4 is not well supported from the data in the test.

Overall this study is good to provide new information to the readers, however, some efforts have to be done to its improvements.

Reviewer 2 Report

The manuscript entitled:”Antibacterial effectiveness of fecal water and in vitro activity of a multistrain probiotic formulation against carbapenem-resistant Gram-negative microorganisms and methicillin-resistant Staphylococcus aureus” was thoroughly reviewed. The results showed that the multistrain probiotics may have decolonization effects on the multi-drug resistant microorganisms in human body. These findings are valuable and supported by the results of this experiment. The manuscript preparation is good but there are some major concerns about the design of this experiment:

There are only 6 individuals used for testing the multistrain probiotics. As it is mentioned there are four treatment groups. So this means only two individuals for each treatment group. This is not enough to build-up strong conclusions. The experiment period seems a bit short. Seven days might not be enough to investigate the somatic changes

The title seems a little long please modify.

Abstract: This part is not well-written. Different treatments should be clearly explained here. For example, what are CRpPa, CR-Ab, CR-Kp, and MRSA? Also the number individuals used for each treatment should be stated.

Line 20: isn’t it better to abbreviate whole probiotic as WP instead of PW?

Line 24: I think you need to put microorganisms after MDR

Line 38: It is better to mention the full name of E. coli and K. pneumonia for the first time

Line 43: … and antibiotics as the drugs…

Line 43: Delete comma after trials

Line 68: The number of individual subjects should be stated here

Round 2

Reviewer 1 Report

This draft can be accepted for publication.

Reviewer 2 Report

no comments